# Giant Pulmonary Artery Thrombotic Material, Due to Chronic Thromboembolic Pulmonary Hypertension, Mimics Pulmonary Artery Sarcoma

**DOI:** 10.3390/medicina57090992

**Published:** 2021-09-20

**Authors:** Helen Triantafyllidi, Dimitrios Iordanidis, Aikaterini Mpahara, Maria Mademli, Dionyssia Birmpa, Stylianos Argentos, Dimitrios Benas, Paraskevi Trivilou, Konstantinos Anagnostopoulos, Eckhard Mayer

**Affiliations:** 12nd Department of Cardiology, Medical School, ATTIKON Hospital, University of Athens, 12462 Athens, Greece; iordanidisdimitrios@yahoo.gr (D.I.); katerinampahara@hotmail.com (A.M.); dbirba@gmail.com (D.B.); dimitriosbenas@gmail.com (D.B.); ptrivilou@yahoo.gr (P.T.); 22nd Department of Radiology, Medical School, ATTIKON Hospital, University of Athens, 12462 Athens, Greece; mariamademli@yahoo.gr (M.M.); steliosargentos@yahoo.gr (S.A.); 3Center for Experimental Surgery, Clinical & Translational Research, Biomedical Research Foundation Academy of Athens, 11527 Athens, Greece; cdanagnostopoulos@bioacademy.gr; 4Heart and Thorax Center, Department of Thoracic Surgery, University of Giessen, Campus Kerckhoff, 61231 Bad Nauheim, Germany; e.mayer@kerckhoff-klinik.de

**Keywords:** chronic thromboembolic pulmonary hypertension, pulmonary artery sarcoma, pulmonary endarterectomy

## Abstract

In this article, we present the case of a 38-year-old female who suffered from serious respiratory distress. After an extensive pulmonary artery imaging diagnostic work-up (CTPA, MRA and PET), we were unable to differentiate between chronic thromboembolic pulmonary hypertension (CTEPH) vs. pulmonary artery sarcoma (PAS) due to extensive filling defects and extraluminal findings. Although surgery was postponed for nine months due to the COVID-19 pandemic, CTEPH diagnosis, due to a high-thrombus burden, was finally confirmed after pulmonary endarterectomy (PEA). Conclusively, imaging findings of rare cases of CTEPH might mimic PAS and the surgical removal of the lesion are both needed for a final diagnosis. *What is Already Known about This Topic?* Pulmonary artery sarcoma (PAS) is a rare but aggressive malignancy, which originates from the intimal layer of the pulmonary artery (PA); Chronic thromboembolic pulmonary hypertension (CTEPH) is based on chronic, organized flow-limiting thrombi inside PA circulation and subsequent pulmonary hypertension. *What Does This Study Contribute?* Since radiological findings of CTEPH cases might rarely mimic PAS, pulmonary artery endarterectomy and subsequent histopathologic study are needed for a final diagnosis.

## 1. Introduction

Several pulmonary vessel diseases (chronic thromboembolic pulmonary hypertension (CTEPH), pulmonary artery sarcoma (PAS), large vessel pulmonary artery vasculitis, congenital pulmonary artery branch stenosis) are characterized by multiple filling defects inside pulmonary circulation. In rare cases, it maybe difficult to differentiate between those above-mentioned diseases, despite the use of several advanced imaging techniques (CTPA, MRA, PET).

## 2. Case Report

A 38-year-old female, previous smoker with a known medical history of idiopathic thrombocytopenic purpura (ITP) and subsequent splenectomy 7 years ago was referred to our Cardiology Department for an extensive evaluation and treatment after an unprovoked pulmonary embolism a month prior. At the time of admission, she was under continuous oxygen treatment due to serious respiratory distress (80% O_2_ saturation in room air) and had a severely dysfunctional right heart ventricle (RVSP = 110 mmHg estimated by echocardiography) and NYHA IV. Computed tomography of the pulmonary artery (CTPA) revealed multiple filling defects in pulmonary artery (PA) circulation, mainly in the right main branch (RPA) as well as in lobar and segmental branches for the right upper lobe. However, the Radiology Department consultation led to the suspicion of an RPA sarcoma instead of thrombotic intrapulmonary artery material, since “the filling defect had lobulated margins and expanded beyond the margins of the vessel wall while there was a faint indication of an intralesional feeding vessel.” Systemic lupus erythymatosus (SLE) was diagnosed, while the patient also tested positive for lupus anticoagulant (LA) and anticardiolipin antibodies (aCL). 

In order to differentiate RPA-derived sarcoma vs. giant intrapulmonary artery thrombotic material, we proceeded to PA magnetic resonance angiography (MRA) and positron emission tomography (PET). However, both tests were inconclusive regarding differential diagnosis between CTEPH vs. PAS since “the lesion expanded beyond the margins of the vessel wall (MRA, PET) having a low signal intensity similar to that of the muscles on T1WI, heterogeneous high signal intensity on T2WI with fat saturation and progressively increasing enhancement after Gadolinium injection”; see Figure 1 (panels A(a,b,c) and B). The patient was discharged under continuous oxygen supply, coumarin anticoagulants, hydroxychloroquine (Plaquenil) 200 mg ×2 and methylprednisolone (Medrol) 16 mg ×1. She was scheduled for an immediate thoracotomy instead of a transcatheter pulmonary biopsy, which has possible complications (i.e., pneumothorax, pulmonary hemorrhage, hemothorax, intrapulmonary hemorrhage, pulmonary artery rupture). Unfortunately, the COVID-19 pandemic postponed the surgery for several months and the patient progressively deteriorated while on anticoagulants. She was re-evaluated every month at the Outpatient Pulmonary Hypertension Clinic. The patient remained with NYHA IV functional status, she needed continuous oxygen therapy (24h/day) and increased doses of furosemide (~500mg/day) and she tried to keep INR ~3 while being again hospitalized (twice) in order to receive intravenous furosemide. Nine months post-initial admission of the patient in our Cardiology Department, newer MRI imaging findings were unchanged, and we finally managed to reschedule the patient for thoracotomy. The right heart catheterization (RHC), just prior to surgery, revealed severe pre-capillary pulmonary hypertension (PH) (mean pulmonary artery pressure (mPAP) = 57 mmHg, pulmonary vascular resistance (PVR) = 15 Wood Units, cardiac index (CI) = 1.59 l/min/m^2^, pulmonary artery wedge pressure (PCWP) = 13 mmHg). During pulmonary endarterectomy (PEA), RPA was found to be severely calcified, fresh clots were revealed and extracted along with multiple partially organized adherent thrombi inside pulmonary circulation, Figure 1 (Panel C). However, no evidence of vasculitis or malignancy was found. The final diagnosis of chronic thromboembolic pulmonary hypertension (CTEPH) due to severe thromboembolism and incomplete thrombotic material lysis was confirmed. At 6 months post-operative re-evaluation, the patient remainedin NYHA I functional status with no signs of pulmonary hypertension at RHC (mPAP = 24 mmHg, PVR = 1.08 Wood Units, CI = 2.98 l/min/m^2^). The antiphospholipid syndrome diagnosis was not finally confirmed.

## 3. Discussion

Pulmonary artery sarcoma (PAS), a rare but aggressive malignancy, originates from the mesenchymal cells of the intimal layer of the pulmonary artery (PA). It usually occupies the main PA branches while it sometimes expands extraluminally [1]. Due to similarities in the clinical features (signs of pulmonary hypertension and right-heart failure) and imaging findings on CTPA (a heterogeneously enhancing low-attenuation filling defect occupying the entire diameter of the PA branches or even expanding extravasculary), PAS and CTEPH are often difficult to distinguish between [1,2]. Similarly, the presence of multiple mismatched perfusion defects on the V/Q scan can mimic CTEPH, leading to delays in the diagnosis and treatment of PAS [3]. On PET scans, PAS usually demonstrates increased FDG (18 F-fluorodeoxyglucose) uptake, whereas CTEPH does not usually show PET avidity [2]. On MRI, PAS demonstrates areas of restricted diffusion, due to increased cellularity and delayed heterogeneous enhancement, not usually seen in CTEPH [4]. Finally, while PEA is usually curative for CTEPH, it just relieves the obstruction and improves symptoms in PAS [2,5].

CTEPH diagnosis is based on chronic, organized flow-limiting thrombi with PA circulation, demonstrated by ventilation/perfusion (V′/Q′) scanning, CTPA or pulmonary angiography for at least 3 months under effective anticoagulation, while RHC confirms PH. PEA is the treatment of choice for operable CTEPH, since the majority of patients experience substantial relief from symptoms and near-normalization of hemodynamics [5,6].

In our patient, radiological findings (CTPA, PET, MRI) led us to the suspicion of a PAS diagnosis, due to multiple filling defects inside pulmonary circulation and a non-homogenous RPA filling defect with lobulated margins, which seemed to expand beyond the margins of the vessel wall, accompanied by a feeding vessel. Although PEA was the surgical method of choice for both CTEPH and PAS, surgery was postponed for 9 months due to the COVID-19 pandemic. Thankfully, CTEPH was finally diagnosed, instead of PAS, after the surgical removal of the thrombotic lesions by PEA. The patient fully recovered after surgery and remains atNYHA status I 9 months postoperatively. 

Conclusively, rare cases of CTEPH maymimic PAS. Surgical removal of the lesion after PEA and histopathological study are needed for a final diagnosis.

## Figures and Tables

**Figure 1 medicina-57-00992-f001:**
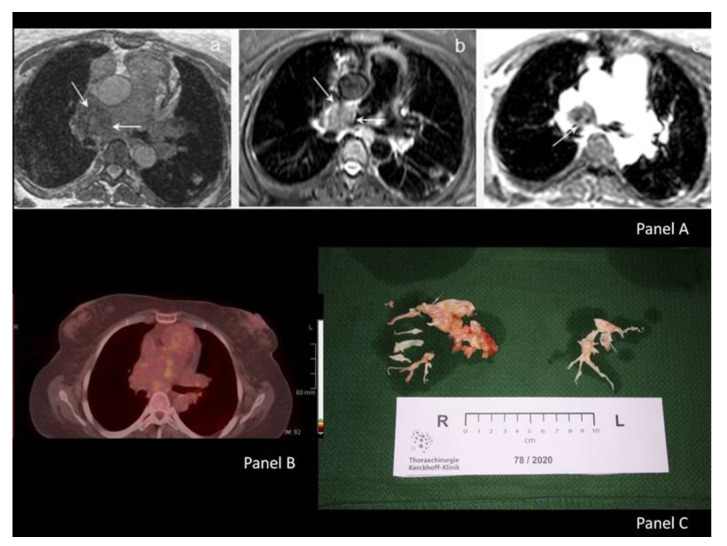
(**Panel A**) In MRA, the lesion expanded beyond the margins of the vessel wall having: a low signal similar to that of the muscles (T1WI) (**a**), an heterogeneous high signal intensity (T2W2) with fat saturation (**b**) and finally progressive increase in the enhancement of the lesion during post GD injection (instance from MR perfusion imaging) (**c**); (**Panel B**) In PET, excessive tissue emanating from the RPA and extending towards the hilum and the sub-carinal region with inhomogeneous FDG distribution (SUVmax: 3.1); (**Panel C**) Surgical specimen with fresh clots along with multiple partially organized adherent thrombi.

## Data Availability

Data is contained within the article.

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
