# Peer review of "Giant Pulmonary Artery Thrombotic Material, Due to Chronic Thromboembolic Pulmonary Hypertension, Mimics Pulmonary Artery Sarcoma"

_medicina, 2021, doi:10.3390/medicina57090992_

Round 1
Reviewer 1 Report
Thank you very much for the opportunity to review this article which I read with great interest.
It is interesting case report in which the diagnoses of CTEPH and PAS were carefully discussed.
I have some concerns or comments as follows.
Major
- If the patient underwent FDG-PET (fluorodeoxyglucose positron emission tomography), please clarify the value of FDG uptake (maximum standardized uptake value [SUVmax]).
- Did the authors consider to perform trans catheter pulmonary biopsy? The pathological examination could reveal that the mass was CTEPH or sarcoma. The surgical approaches for the treatment of these diseases might be different. Pulmonary endarterectomy with circulatory arrest for CTEPH or CTED, and pneumonectomy and lymph node dissection for pulmonary angiosarcoma.
Minor
- Page 2, line 71. ‘pulmonary arterial hypertension (PAH)’ might be ‘pulmonary hypertension (PH)’.
Author Response
Reviewer #1:
Thank you very much for the opportunity to review this article which I read with great interest. It is interesting case report in which the diagnoses of CTEPH and PAS were carefully discussed. I have some concerns or comments as follows.
Major
- If the patient underwent FDG-PET (fluorodeoxyglucose positron emission tomography), please clarify the value of FDG uptake (maximum standardized uptake value [SUVmax]).
â–ş Answer: SUVmax: 3.1. We added the value in the legend of Panel B of Figure 1.
- Did the authors consider to perform trans catheter pulmonary biopsy? The pathological examination could reveal that the mass was CTEPH or sarcoma. The surgical approaches for the treatment of these diseases might be different. Pulmonary endarterectomy with circulatory arrest for CTEPH or CTED, and pneumonectomy and lymph node dissection for pulmonary angiosarcoma.
â–ş Answer: You are totally right. However, we decided to plan the patient for thoracotomy instead of a trans catheter pulmonary biopsy and its complications (intrapulmonary hemorrhage, pulmonary artery rupture). Unfortunately, COVID-19 pandemic postponed the surgery for several months while the patient progressively and totally deteriorated. Luckily, we finally managed to re-schedule the patient for thoracotomy (text added).
Minor
- Page 2, line 71. ‘pulmonary arterial hypertension (PAH)’ might be ‘pulmonary hypertension (PH)’.
â–şAnswer: We corrected it.
Reviewer 2 Report
Thank you for this interesting artcile. I have some suggestions for this manuscript:
- In Abstract: The case of a 38-year-old female suffered from serious respiratory distress is presented.- I would rewrite it as: This is a case of a 38 year old female who suffered from serious respiratory distress. Or in this article, we have presented........
- idiopathic thrombopenic purple-what is this diagnosis, is it same as ITP?
- What was the treatment given when the patient waited for the surgery. It was mentioned that she was discharged with O2 and anticoagulation, but more information about the clinical course/follow up will be better.
- It was mentioned that the patient was diagnosed with Systemic lupus erythymatosus (SLE) and the patient tested also positive for lupus anticoagulant (LA) and anticardiolipin antibodies (aCL).-was the patient diagnosed with antiphospholipid syndrome? Was it a new diagnosis as a part of the evaluation of the pulmonary embolism? Was there any other tests done to assess the hypercoagulable status of the patient?
- It would be interesting to know what will be the role of V/Q scan in such clinical scenario to differentiate between CTEPH and PAS
- It would be better if we can get better quality images
Author Response
Reviewer #2:
Thank you for this interesting article. I have some suggestions for this manuscript:
- In Abstract: The case of a 38-year-old female suffered from serious respiratory distress is presented.- I would rewrite it as: This is a case of a 38 year old female who suffered from serious respiratory distress. Or in this article, we have presented........
â–şAnswer: We rephrased the 1st sentence.
- Idiopathic thrombopenic purple-what is this diagnosis, is it same as ITP?
â–şAnswer: You are quite right, it should be written as idiopathic thrombocytopenic purpura (ITP) which is the same disease with immune thrombocytopenic purpura (ITP).
- What was the treatment given when the patient waited for the surgery. It was mentioned that she was discharged with O2 and anticoagulation, but more information about the clinical course/follow up will be better.
â–şAnswer: We agree and we added a short text.
- It was mentioned that the patient was diagnosed with Systemic lupus erythymatosus (SLE) and the patient tested also positive for lupus anticoagulant (LA) and anticardiolipin antibodies (aCL).-was the patient diagnosed with antiphospholipid syndrome? Was it a new diagnosis as a part of the evaluation of the pulmonary embolism? Was there any other tests done to assess the hypercoagulable status of the patient?
â–şAnswer: Indeed, the patient was diagnosed with a possible antiphospholipid syndrome during the pre-surgical period evaluation of any thrombophilia disorder (acute phase). However, at the six months post-operative evaluation, the antiphospholipid syndrome diagnosis was not confirmed.
- It would be interesting to know what will be the role of V/Q scan in such clinical scenario to differentiate between CTEPH and PAS
â–şAnswer: In PAS, the presence of multiple mismatched perfusion defects on a VQ scan and pulmonary arterial filling defects on a CTPA can mimic CTEPH. PAS is frequently misdiagnosed leading to delays in diagnosis and treatment (text and reference added).
- It would be better if we can get better quality images
â–şAnswer: Unfortunately, this is the best we have despite our efforts